# Relevance of Biomarkers in Serum vs. Synovial Fluid in Patients with Knee Osteoarthritis

**DOI:** 10.3390/ijms24119483

**Published:** 2023-05-30

**Authors:** Stefania Kalogera, Mylène P. Jansen, Anne-Christine Bay-Jensen, Peder Frederiksen, Morten A. Karsdal, Christian S. Thudium, Simon C. Mastbergen

**Affiliations:** 1Nordic Bioscience, Biomarkers and Research, 2730 Herlev, Denmark; kalogera.stefi@gmail.com (S.K.); acbj@nordicbio.com (A.-C.B.-J.); pef@nordicbio.com (P.F.); mk@nordicbio.com (M.A.K.); 2Department of Drug Design and Pharmacology, Copenhagen University, 1165 Copenhagen, Denmark; 3Department of Rheumatology & Clinical Immunology, UMC Utrecht, Utrecht University, 3584 CX Utrecht, The Netherlands; m.p.jansen-36@umcutrecht.nl (M.P.J.); s.mastbergen@umcutrecht.nl (S.C.M.); 4Regenerative Medicine Center, Utrecht University, 3584 CS Utrecht, The Netherlands

**Keywords:** osteoarthritis, biomarkers, synovial fluid, pain, radiographical outcomes, extracellular matrix

## Abstract

The association between structural changes and pain sensation in osteoarthritis (OA) remains unclear. Joint deterioration in OA leads to the release of protein fragments that can either systemically (serum) or locally (synovial fluid; SF) be targeted as biomarkers and describe structural changes and potentially pain. Biomarkers of collagen type I (C1M), type II (C2M), type III (C3M), type X (C10C), and aggrecan (ARGS) degradation were measured in the serum and SF of knee OA patients. Spearman’s rank correlation was used to assess the correlation of the biomarkers’ levels between serum and SF. Linear regression adjusted for confounders was used to evaluate the associations between the biomarkers’ levels and clinical outcomes. The serum C1M levels were negatively associated with subchondral bone density. The serum C2M levels were negatively associated with KL grade and positively associated with minimum joint space width (minJSW). The C10C levels in SF were negatively associated with minJSW and positively associated with KL grade and osteophyte area. Lastly, the serum C2M and C3M levels were negatively associated with pain outcomes. Most of the biomarkers seemed to mainly be associated with structural outcomes. The overall biomarkers of extracellular matrix (ECM) remodeling in serum and SF may provide different information and reflect different pathogenic processes.

## 1. Introduction

Osteoarthritis (OA) is the most common form of arthritis and is characterized by chronic pain and disability [1]. The main hallmarks of OA include cartilage degradation, osteophyte formation, subchondral bone remodeling, and inflammation of the synovium. Joint pain is the most common clinical OA manifestation; however, the association between structural changes and pain is still not completely understood. There is a discordance between radiographic damage and pain, as up to 40% of OA patients with radiographic OA report no pain [2]. Thus, there is a clear need for biological tools that can describe the underlying processes and aid in understanding the link between structural pathology and pain development [3].

OA is a disease with high tissue turnover involving alterations of the extracellular matrix (ECM) of the joint. These alterations are driven by the upregulation of inflammatory cytokines and proteolytic enzymes. The ECM of the joint largely consists of collagens, proteoglycans, and other proteins such as fibronectin, elastin, and laminin [4]. The continuous changes to the ECM in OA result in the release of neoepitope fragments into the circulation. These fragments derive from the proteolytic cleavage of ECM proteins by specific proteases, and they can be quantified by immunoassays and used as biomarkers [5]. These markers reflect cartilage degradation biomarkers, bone resorption biomarkers, and connective tissue remodeling and inflammation biomarkers. These markers can be either systemically (in serum) or locally (in synovial fluid; SF) measured [6]. Since SF covers the entire joint cavity, it can be an important diagnostic tool for assessing tissue turnover, as it is a more direct or local measure than blood.

In this study, we investigated the levels of the following biomarkers of ECM remodeling: type I (C1M), type II (C2M), type III (C3M), type X (C10C) collagen, and aggrecan (ARGS) degradation in the serum and SF of OA patients. These markers were evaluated before in serum but have not yet been combined with the SF of OA patients. Collagen remodeling markers were associated with distinct OA characteristics [7]. C1M, which is a matrix metalloproteinase (MMP)-2-, -9-, and -13-mediated degradation fragment of type I collagen [8], was correlated with synovitis and pain in the serum of end-stage knee OA patients [9]. C2M is an MMP-9-generated degradation fragment of type II collagen, and it was found to be elevated in the serum of patients with mild and severe OA compared to healthy individuals [10]. In SF, collagen type II formation markers were previously assessed, and they were associated with the treatment response to sprifermin, which is a cartilage anabolic drug [11]. This highlights the importance of SF biomarkers to assess responses to local therapeutic interventions. Kjelgaard-Petersen et al. stated that C3M, which is another collagen biomarker, is an MMP-9-generated fragment of type III collagen reflecting tissue degradation, and it is released from human OA synovial explants after pro-inflammatory stimuli [8]. Serum C3M levels were negatively associated with knee symptoms assessed by Western Ontario and McMaster Universities Osteoarthritis Index (WOMAC) scores in patients with symptomatic knee OA [12]. However, in another study, serum C3M levels were found to predict the risk of developing painful radiographic knee OA within 10 years [13]. C10C is a cathepsin-K-mediated fragment of type X collagen reflecting chondrocyte activity, and it has been found to be upregulated in the serum of OA patients with Kellgren–Lawrence (KL) grade 2 compared to healthy individuals as well as in patients with high C-reactive protein (CRP) levels [14]. Aggrecan is a major component of the ECM of articular cartilage and is part of the cartilage metabolism during OA. Fragments of aggrecan can be generated by a disintegrin and metalloproteinase with thrombospondin motifs (ADAMTS)-5, which is upregulated and activated in OA [15]. ARGS is a fragment resulting from the proteolytic cleavage form of aggrecan by ADAMTS4/5. ARGS was found to be associated with disease activity in OA, as patients with low serum levels at baseline were more likely to progress and to respond to treatment [16]. Other aggrecan neoepitope fragments were previously assessed in SF and were correlated with WOMAC pain scores [17]. Overall, the literature analysis shows that in SF the most studied biomarkers were the inflammatory biomarkers, such as different interleukins or proteinases [18]. However, the soluble tissue remodeling biomarkers that were described above could be useful tools to monitor structural and pain progression and objective tools to evaluate treatment responses. Assessing the levels of a panel of ECM biomarkers in two different matrices could reflect the dynamic and quantitative changes in joint remodeling. 

The aims of this study were to (1) investigate the relationship between the levels of the ECM degradation biomarkers in serum and the same biomarkers’ levels in SF and (2) investigate the associations between the levels of the aforementioned biomarkers and the clinical outcomes in OA serum and SF.

## 2. Results

### 2.1. Patient Characteristics

A total of 31 individuals with knee OA, consisting of 19 females and 12 males with an average age of 68.4 ± 6.7 years, were included in this study. The demographics and clinical characteristics from month 24 are presented in Table 1. The levels of the biomarkers in both serum and SF were within the respected measurement range for each assay. Some of the demographics and clinical characteristics differed between the subset and the full dataset. The OA individuals in the subset had a higher Knee Injury and Osteoarthritis Outcome Score (KOOS) pain score (*p* < 0.001), which translated to less pain, compared to the full dataset, since the KOOS pain scale is inverted. Similarly, the Numeric Scale Score (NRS) and Intermittent and Constant OA Pain (ICOAP) pain scores were lower in the OA individuals of the subset compared to the full dataset (*p* = 0.016 and *p* = 0.009, respectively). No other differences were observed in the demographics and the clinical characteristics from month 24 between the two groups (Table 1).

### 2.2. Biomarkers’ Levels in Serum and SF

The C1M, C2M, C3M, and C10C levels were significantly higher (*p* < 0.0001 in all markers) in serum than in SF, while the ARGS levels were higher in SF (Figure 1). The C2M and C3M levels in serum were weakly to moderately correlated with the C2M (ρ = 0.46) and C3M (ρ = 0.38) levels in SF, respectively. No significant correlations were observed between the remaining biomarkers’ levels in serum and their respective levels in SF (Figure 2).

### 2.3. Relationship between Biomarkers’ Levels and Structural Outcomes in Serum and in SF

The associations between the biomarkers’ levels in serum and SF, with the structural outcomes, are summarized in Table 2. The C2M levels in serum were negatively associated with KL grade (relative difference = 0.903, *p* = 0.049), meaning that for each unit increase in the KL grade, C2M levels in serum are decreasing by 9.7% and positively associated with the minimum joint space width (minJSW) (relative difference = 1.131, *p* = 0.007), meaning that for each mm increase in the joint space width the expected C2M levels are increasing by 13.1%. On the other hand, C10C levels in SF were negatively associated with minJSW (relative difference = 0.837, *p* = 0.011), meaning that for one mm increase in the joint space width expected C10C levels are decreasing by 16.3%. In SF, C10C levels were associated with osteophyte area (relative difference = 1.011, *p* = 0.001), so for one mm^2^ increase in the osteophyte area, expected C10C levels in SF are increasing by 1.1%. Furthermore, C10C levels in SF were associated with the KL grade (relative difference = 1.256, *p =* 0.004), so for each unit increase in the KL grade, C10C levels in SF are increasing by 5.6%. Lastly, serum C1M levels were negatively associated with subchondral bone density (relative difference = 0.938, *p =* 0.001), meaning that for one mm increase in the subchondral bone density expected C1M levels decrease by 6.2%. 

### 2.4. Relationship between Biomarkers’ Levels and Pain Outcomes in Serum and in SF

The associations between biomarkers’ levels in serum and SF with pain outcomes are summarized in Table 3. C2M levels in serum were positively associated with KOOS pain score (relative difference = 1.007, *p =* 0.049), which is translated to negative association since KOOS pain scale is inverted. This association means that for one unit increase in the KOOS pain score, C2M levels in serum are increasing 0.7%. C3M levels in serum were found to be negatively correlated with the outcome of the constant ICOAP pain (relative difference = 0.981, *p =* 0.020), meaning that for each unit increase in the ICOAP score, C10C levels decrease by 1.9 %. The other biomarkers and pain measures showed no statistically significant correlations. 

## 3. Discussion

In our effort toward a better understanding of OA pathological mechanisms and their contribution to pain, in this study, the biomarkers of tissue turnover were measured in both serum and SF, and further associations with clinical outcomes for both matrices were investigated. The main findings of this study were (1) all the markers were both detectable in serum and SF, and some showed a correlation between both compartments, while others did not. This might be explained by their representation of different processes in each matrix. (2) Type I collagen degradation fragment (C1M), type II collagen degradation fragment (C2M), and type X collagen degradation fragment (C10C) were associated with structural outcomes, while (3) type II and type III collagen degradation fragments (C2M and C3M) were correlated with pain outcomes.

### 3.1. Correlation between Biomarkers’ Levels in Serum and SF

Most of the biomarkers’ levels were significantly higher in serum than in SF. It needs to be considered that some of the patients included in this study also had hip OA, which may lead to a higher accumulation of analytes in the systemic circulation, due to the contribution from other joint tissues. The relationship between the biomarkers’ levels in the two different matrices was investigated, and it was found that the concentrations in serum and SF correlate for some biomarkers and vary substantially for others. The C2M and C3M levels were found to correlate between serum and SF, while the C1M, C10C, and ARGS levels did not. The biomarkers that showed a higher correlation between serum and SF may suggest a contribution from systemic processes, whereas the metabolites that show very low correlations reflect either systemic or local processes. Higher concentrations in SF than in serum would suggest a more localized metabolism, thus reflecting joint-specific processes. For example, the ARGS levels were not correlated between serum and SF (ρ = −0.10), but the ARGS levels were higher in SF compared to serum (Figure 1), suggesting that ARGS is a marker that reflects processes in the joint. This variation between serum and SF is also in alignment with data from Zhang et al., who investigated the relationship between plasma and SF metabolite concentrations in OA patients, concluding that the correlation between the two matrices was modest [19]. Similarly, proteomic analyses of plasma and SF for juvenile arthritis revealed significant differences in a subset of 30 proteins, leading to the conclusion that SF was clearly distinguishable from plasma [20]. 

These findings suggest that biomarkers may have different concentrations and reflect different processes, depending on whether they are measured systemically in the circulation or locally in SF.

### 3.2. Relationship between Biomarkers’ Levels and Structural Outcomes in Both Serum and SF

Cartilage degradation is one of the main hallmarks in OA, and type II collagen is the main component of cartilage. C2M, an MMP-derived fragment of type II collagen, reflects cartilage degeneration and was found to be elevated in patients with radiographic knee OA. In this study, we found that the C2M levels were positively correlated with minJSW, which is a measure of cartilage thickness. Since minJSW is a surrogate measure of cartilage thickness, one would hypothesize that patients with a higher minJSW would have less cartilage degradation and consequently lower C2M levels. However, the minJSW measurement reflects a one-dimensional thickness measured from one specific location of the joint, and it may also include other tissues such as the menisci [21]. A correlation analysis between cartilage thickness and minJSW from previous studies showed some variation between the two parameters, which was explained by variation in the cartilage thickness [22,23]. The C2M levels in serum were also found to be negatively correlated with KL grade. Several collagen type II metabolites were found to be positively associated with OA severity [10,24]. However, due to some inconsistencies in the outcomes, it was suggested that panels of collagen type II biomarkers may be a better choice for reflecting radiographical outcomes [25].

On the other hand, C10C, a marker that reflects chondrocyte hypertrophy, was found to be negatively correlated with minJSW in SF, meaning that patients with a higher minJSW would have lower C10C levels, and vice versa. Considering that chondrocyte hypertrophy plays a crucial role in OA progression, by enhancing protease-mediated cartilage degradation, as shown by P M van der Kraan et al. [26], then patients with a lower minJSW, reflecting higher cartilage degradation, are expected to have higher C10C levels.

Osteophyte formation is, in addition to joint space narrowing, one of the main radiographic features of OA. Although osteophyte formation and cartilage degeneration do not completely correlate, Boegård et al. showed that cartilage degradation confirmed by joint space narrowing is associated with the presence of osteophytes [27]. These results could support this study’s finding: C10C levels, which reflect chondrocyte hypertrophy and subsequent cartilage degradation, were found to be correlated with the osteophyte area in SF. 

Furthermore, the C10C levels in SF were positively correlated with disease severity measured by the KL grade. This finding is in alignment with previous data published by He et al., which showed that patients with higher KL grades had significantly higher C10C levels, indicating that chondrocyte hypertrophy is correlated with OA severity [14]. Another type X collagen neo-epitope biomarker showed potential diagnostic value for knee OA patients [28].

Lastly, the C1M levels in serum were negatively associated with subchondral bone density. C1M is a marker reflecting the degradation of type I collagen, a major component of bone, so a higher subchondral bone density reflects less degradation and consequently lower C1M levels. It was previously published by Van de Stadt et al. that C1M levels are associated with erosive hand OA, which is characterized by bone excessive bone resorption and inadequate bone formation [29]. C1M was also found to be associated with an OA phenotype characterized by high remodeling, high radiographic damage, and high inflammation [30].

### 3.3. Relationship between Biomarkers’ Levels and Pain Outcomes in Both Serum and SF

Few studies described biomarkers in relation to pain outcomes in OA. In this study, serum C3M levels were negatively correlated with constant pain, as determined by the ICOAP questionnaire. Constant pain was described as a continuous aching pain sensation, but with less pain intensity [31]. C3M is a marker released from the synovial membrane under inflammatory conditions. The exact role is unclear, as the literature shows different outcomes. Previous studies showed that serum C3M levels were elevated in OA individuals compared to healthy controls [12]. Additionally, it was demonstrated that C3M levels were associated with less severe knee pain in a cross-sectional study with knee OA patients [12]. However, in another study in patients with knee OA, no association was found between changes in C3M levels and changes in pain over 18 months [32]. Lastly, Petersen et al. found a lack of a clear association between inflammation and pain, showing that C3M levels do not seem to be associated with pain outcomes, pressure pain thresholds, or the visual analogue scale (VAS) [33]. The C2M levels in serum were also negatively associated with KOOS pain. A lack of correlation between C2M levels and pain parameters was previously observed [34]. Since cartilage is not innervated by nerves, pain signaling may result from other joint tissues rather than cartilage itself [35]. 

All these results, together with our findings, highlight the observed discordance between radiographic damage and pain observed in many OA studies, as some markers correlate with structural outcomes but not with pain outcomes, and vice versa. This discordance might be due to the complexity of the different OA pain phenotypes, where local joint damage, for example, is not always the main cause of pain, which is usually the case for patients with nociceptive pain. Furthermore, pain sensation varies between individuals and is affected by other factors, such as biological, physiological, and psychosocial factors [36]. 

Another point that should be highlighted is that the subset with the 31 individuals had lower pain scores when compared to the full cohort. This might be one of the reasons for the limited associations that were found between the biomarkers’ levels and the pain outcomes. Despite the fact that everypatient in the University Medical Center Utrecht (UMCU) was asked to participate in this substudy for the collection of synovial fluid, unintentional selection bias might have occurred toward patients who are experiencing less pain and are more open to an intra-articular punction. 

Another limitation of this study is the variability among the measurements of the biomarkers. All serum markers were measured at the Clinical Lab at Nordic Bioscience, while the SF measurements were all performed at the UMCU, and the respective photometers were used for the colorimetric and the chemiluminescence assays. This can introduce some variation among the different measurements, which can affect the comparisons between the two matrices. It needs to be considered that the biomarker assays were technically validated and optimized for serum as the intended matrix. This means that all parameters, such as the lower limit of quantification and the minimum required dilution, should be adjusted for the SF, which is a different matrix. Pretreatment of SF is an additional confounder that could affect the sensitivity and precision of the results. In this study, the SF samples were collected and centrifuged based on the procedures of the lab at the UMCU. Lastly, this study is based on an exploratory analysis, without any power or sensitivity analysis. Further plans could include more OA individuals, as, obviously, the small number of patients used in this study is a clear limitation.

## 4. Materials and Methods

### 4.1. Study Population

A subset of 31 OA patients was recruited from the Innovative Medicines Initiative Applied Public-Private Research enabling OsteoArthritis Clinical Headway (IMI-APPROACH) clinical cohort [37]. IMI-APPROACH is a prospective study consisting of 297 individuals with tibiofemoral OA, according to the classification criteria of the American College of Rheumatology (ACR) [38]. Patients were included in the cohort based on their likelihood of progression. Machine learning tools were used to separate patients with a higher likelihood of progression based on age, minJSW, and KOOS pain. A more detailed characterization of the cohort was described by van Helvoort et al. [37]. Several study procedures were performed, including knee radiographs, pain questionnaires, and serum collection. An index knee was defined for each patient according to the ACR clinical criteria [38]. In case both knees fulfilled the criteria, then the most painful knee was selected as the index knee. In a subset of 31 knee OA patients, all included from UMCU, additional SF samples were obtained from the index knee only at month 24. As such, all analyses in the current study include data from month 24. The 31 individuals included patients from the UMCU who gave their consent for SF collection. Ultrasound was used to check if aspiration would be possible, and, in this case, SF was aspirated. No other specific selection criteria were used for the identification of the participants. The subset and the full dataset of month 24, which included 247 individuals, were compared for differences in the demographics and clinical parameters at month 24. The study was conducted in compliance with the protocol, Good Clinical Practice, the Declaration of Helsinki, and the applicable ethical and legal regulatory requirements (for all countries involved). All patients gave written informed consent to participate in the IMI-APPROACH cohort and additionally for SF aspirations (NL61405.041.17 v03; METC-protocol 17-440/M). 

### 4.2. Pain Evaluation

Pain outcomes from multiple questionnaires were used in this study to investigate any correlations between them and the biomarkers’ levels. The pain subscale of the KOOS) was used to evaluate knee pain [39]. The pain questionnaire consists of nine questions, and each question is scored on a 5-point scale normalized from 0 to 100, with 100 reflecting no pain. NRS; 0-1 with 0 no pain for the index knee reflecting pain was also used [40]. Lastly, the ICOAP questionnaire was used to evaluate both constant and intermittent pain. This questionnaire consists of five questions for constant pain and six questions for intermittent pain, scored on a 5-point scale and then normalized to a 100-point scale, with 0 reflecting no pain and 100 reflecting the highest pain [31]. 

### 4.3. Structural Evaluation

Radiographs of the index knee were performed at month 24, from which the KL grade was determined, and minJSW (mm), total osteophyte area in the joint (mm^2^), and subchondral bone density (mm) were measured using Knee Images Digital Analysis (KIDA) [41]. 

### 4.4. Sample Processing

#### 4.4.1. SF Samples

SF samples were collected by needle aspiration with an 18-gauge needle under ultrasound guidance from OA patients at the UMCU who were visiting for their month 24 visit of the APPROACH study. A maximum of 2 mL of SF was aspirated by needle from the index knee within 1 h, and samples were subsequently centrifuged for 10 min at 2300× *g*. Samples were checked and scored for blood contamination before and after centrifugation. None of the samples were contaminated after centrifugation. Supernatants were stored in 200 μL aliquots in cryovials at −80 °C in monitored freezers. 

#### 4.4.2. Serum Samples

Serum samples were collected under the IMI-APPROACH cohort protocols at their month 24 visit. All samples were stored at −80 °C until analysis. 

### 4.5. Biomarker Assays

Biomarkers were measured in serum and SF from patients by technically validated enzyme-linked immunosorbent assays (ELISAs). The assays were developed based on the Protein FingerPrint™ technology, a technology for quantifying neo-epitopes, which are epitopes that are generated after the cleavage of a protein by certain proteases. All biomarkers were technically validated assays and were run in accordance with the instructions of the manufacturer. The markers included in this study and the respective references to their technical papers are listed in Table 4.

Briefly, streptavidin-coated 96-well plates were coated with respective biotinylated antigens for 30 min at 20 °C and subsequently washed 5 times in washing buffer. Appropriate controls, standards serially diluted in assay buffer, and samples were incubated with horseradish peroxidase-conjugated monoclonal antibodies for 1 h at 20 °C (C3M) or 20 h at 4 °C (C1M, C2M, C10C, and ARGS), under 300 rpm shaking. Subsequently, 100 μL of either peroxidase-conjugated or unlabeled antibody were added to the wells, against the respective epitope. Plates were subsequently washed 5 times with washing buffer. For the unlabeled antibody, a secondary horse peroxidase-conjugated labeled antibody was subsequently added and incubated for 1 h at 20 °C, under 300 rpm shaking. 

For the colorimetric assays, tetramethylbenzidine substrate (TMB) substrate was added, and the plates were incubated for 15 min at 20 °C, under 300 rpm shaking. Plates with TMB were stopped by the addition of 1% H_2_SO_4_. A SpectraMax Microplate Reader (Molecular Devices Corporation, Sunnyvale, CA, USA) was used to read the absorbance at 450 nm, with reference set to 650 nm for the TMB assays in serum, and iMark TM Microplate Absorbance Reader (BioRad Laboratories, Inc., Hercules, CA, USA) was used for the SF measurements.

For the chemiluminescence ones, BM Chemiluminescence ELISA Substrate was added to the wells and incubated in dark for 3 min, under 300 rpm shaking. Plates were read with a light emission at 1000 ms and no specific wavelength filter, using SpectraMax Microplate Reader (Molecular Devices Corporation, Sunnyvale, CA, USA) for the serum measurements and CLARIOstar^®^ Plus (BMG LABTECH, Cary, NC, USA) for the SF measurements. The concentrations were calculated by using a 4-parametric standard curve fit model. Biomarker results were approved by fulfilling acceptance requirements of an analytical run. All samples were measured in duplicates. All runs were quality-controlled by two kit controls and three in-house quality controls. Plates were valid only when the standard curve had a coefficient of variance below 15%, samples’ coefficient of variance wase below 20%, and 3 out of the 5 quality controls were between the accepted range. 

## 5. Conclusions

The findings of this study suggest that biochemical markers in serum and SF may provide differing information and reflect different underlying processes. Biomarkers that correlated between both matrices, such as C2M and C3M, may provide information about the joint pathophysiology through serum analysis and may be associated with changes in multiple joints or organs, while ARGS and C10C are likely a reflection of more specific local or systemic processes. Not surprisingly, given their origin in the ECM, most of the biomarkers, including C1M, C2M, and C10C, seemed to mainly be associated with structural outcomes. These exploratory findings further substantiate the necessity of biomarkers as tools for describing OA pathological changes both locally and systemically.

## Figures and Tables

**Figure 1 ijms-24-09483-f001:**
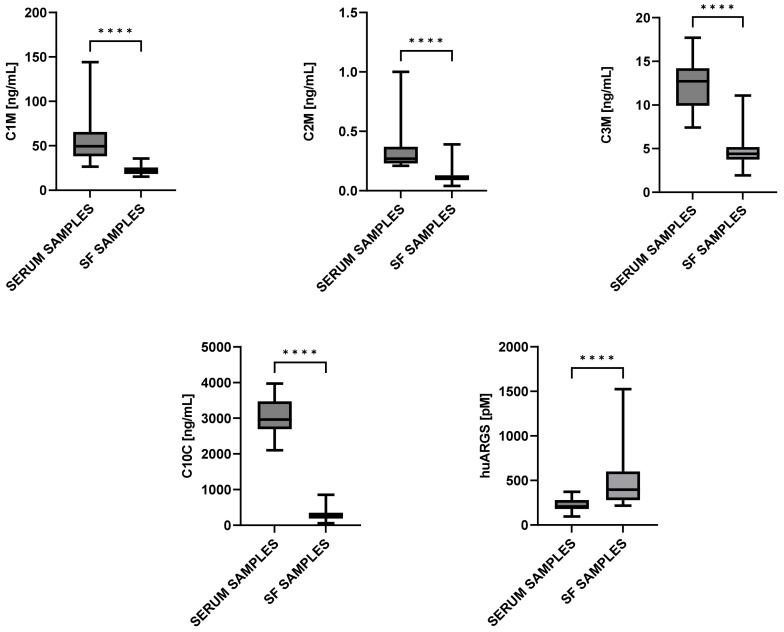
C1M, C2M, C3M, C10C, and ARGS levels in serum and in SF. Data are shown as boxplots showing min to max ± SD. Differences between the two groups were assessed by Wilcoxon paired rank test. Statistical significance is considered as **** = *p* < 0.0001.

**Figure 2 ijms-24-09483-f002:**
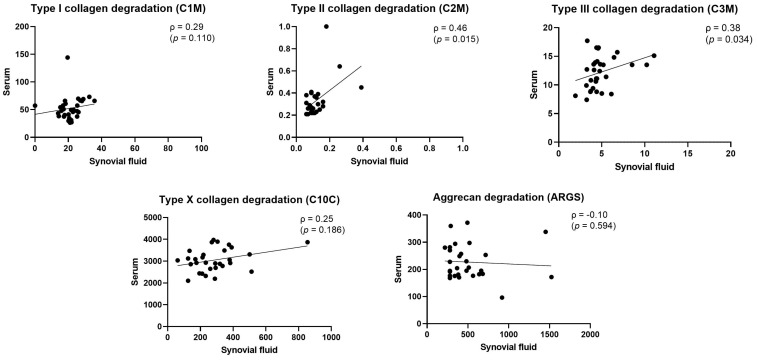
Relationship between log-transformed biomarkers’ levels in SF and serum. Each dot represents a single participant. Linear regression line is shown, and the Spearman correlation coefficient (ρ) shows the strength of the relationship between the two ranked variables, with the respective *p*-value.

**Table 1 ijms-24-09483-t001:** Demographics and clinical characteristics of the full dataset (*n* = 247) and the subset population (*n* = 31). Continuous variables are described as mean (SD), while categorical variables are described as counts (percentages). Differences in demographics and clinical characteristics from month 24 were assessed with Wilcoxon rank test for continuous variables and chi-square test for categorical variables. Significant data are highlighted in bold.

	Full Dataset (*n* = 247)	Subset (*n* = 31)	*p*-Value
Age (SD)	66.5 (7.1)	68.4 (6.7)	0.090
Female, n (%)	176 (71.2)	19 (61.3)	0.082
BMI (SD)	28.3 (5.3)	27.2 (4.5)	0.506
KOOS Pain	63.4 (20.6)	76.2 (18.9)	**<0.001**
NRS Pain	4.4 (2.8)	3.4 (2.8)	**0.016**
ICOAP Pain	26.3 (21.0)	16.6 (17.5)	**0.009**
minJSW (mm)	2.4 (1.2)	2.3 (1.5)	0.924
Osteophyte Area (mm^2^)	26.9 (25.7)	34.8 (29.62)	0.158
Subchondral Bone Density (mm)	30.1 (4.9)	15.3 (7.8)	0.096
KL Grade (%)			
0	38 (15.4)	6 (19.4)	0.537
1	63 (25.5)	7 (22.6)	
2	47 (19.0)	5 (16.1)	
3	73 (29.6)	9 (29.0)	
4	10 (4.0)	4 (12.9)	

**Table 2 ijms-24-09483-t002:** Relative difference in expected C1M, C2M, C3M, C10C and ARGS biomarker levels per one unit increase in KL grade, minJSW, osteophyte area, or subchondral bone density. The respective relative differences, 95% confidence intervals and p-values are shown in the table. Models were adjusted for age, sex, and BMI. Significant data are highlighted in bold.

Biomarkers/ Clinical Variables	KL Grade			MinJSW			Osteophyte Area			Subchondral Bone Density		
	Relative Difference	95% CI	*p*	Relative Difference	95% CI	*p*	Relative Difference	95% CI	*p*	Relative Difference	95% CI	*p*
C1M_Serum	0.955	0.864–1.056	0.378	1.026	0.936–1.124	0.595	0.988	0.975–1.001	0.155	0.938	0.908–0.970	**0.001**
C2M_Serum	0.903	0.819–0.995	**0.049**	1.131	1.041–1.228	**0.007**	0.998	0.994–1.003	0.486	1.014	0.972–1.057	0.532
C3M_Serum	1.011	0.944–1.083	0.756	1.010	0.949–1.075	0.760	1.001	0.998–1.004	0.630	0.994	0.967–1.022	0.671
C10C_Serum	1.030	0.982–1.081	0.231	0.974	0.932–1.017	0.247	1.001	0.999–1.003	0.464	1.011	0.991–1.031	0.287
ARGS_Serum	1.077	0.999–1.161	0.063	0.957	0.891–1.027	0.233	1.003	1.000–1.007	0.055	0.996	0.964–1.028	0.789
C1M_SF	0.984	0.919–1.052	0.637	1.003	0.942–1.068	0.921	1.000	0.996–1.012	0.917	1.004	0.977–1.032	0.778
C2M_SF	0.994	0.86–1.150	0.941	1.048	0.923–1.191	0.475	1.004	1.005–1.043	0.195	1.021	0.966–1.080	0.465
C3M_SF	1.014	0.931–1.104	0.753	0.940	0.873–1.013	0.115	1.000	0.982–1.007	0.869	0.988	0.955–1.023	0.503
C10C_SF	1.256	1.092–1.445	**0.004**	0.837	0.737–0.950	**0.011**	1.011	0.976–1.018	**0.001**	1.044	0.982–1.109	0.180
ARGS_SF	0.889	0.768–1.029	0.129	1.123	0.989–1.274	0.086	0.095	0.986–1.033	0.148	0.990	0.934–1.050	0.752

**Table 3 ijms-24-09483-t003:** Relative difference in expected C1M, C2M, C3M, C10C, and ARGS biomarker levels per one unit increase in KOOS pain, NSR pain, and ICOAP pain score (constant, intermittent, and total). The respective relative differences, 95% confidence intervals, and p-values are shown. Models were adjusted for age, sex, and BMI. Significant data are highlighted in bold.

Biomarkers/ Clinical Variables	KOOS Pain				NRS Pain			ICOAP Pain			
	Relative Difference	95% CI	*p*	Relative Difference		95% CI	*p*		Relative Difference	95% CI	*p*
C1M_Serum	1.002	0.994–1.009	0.671	0.990		0.937–1.046	0.718	Constant	0.997	0.988–1.007	0.584
								Intermittent	1.002	0.994–1.010	0.609
								Total	1.000	0.991–1.009	0.949
C2M_Serum	1.007	1.000–1.015	**0.049**	0.961		0.910–1.016	0.173	Constant	0.991	0.982–1.000	0.059
								Intermittent	0.993	0.985–1.001	0.112
								Total	0.992	0.983–1.000	0.071
C3M_Serum	1.004	0.999–1.009	0.130	0.966		0.932–1.000	0.064	Constant	0.993	0.987–0.999	**0.020**
								Intermittent	0.997	0.992–1.002	0.268
								Total	0.091	0.989–1.001	0.995
C10C_Serum	0.999	0.995–1.002	0.497	1.012		0.985–1.039	0.396	Constant	1.000	0.996–1.005	0.978
								Intermittent	1.001	0.998–1.005	0.482
								Total	1.001	0.997–1.005	0.664
ARGS_Serum	0.999	0.993–1.005	0.793	1.009		0.966–1.054	0.685	Constant	0.999	0.992–1.007	0.867
								Intermittent	1.003	0.997–1.010	0.310
								Total	1.002	0.995–1.009	0.594
C1M_SF	1.000	0.995–1.003	0.996	1.002		0.965–1.039	0.929	Constant	1.000	0.993–1.008	0.941
								Intermittent	0.999	0.993–1.004	0.667
								Total	1.000	0.991–1.001	0.949
C2M_SF	1.004	0.988–1.009	0.396	0.953		0.887–1.025	0.209	Constant	0.993	0.980–1.006	0.292
								Intermittent	0.991	0.981–1.002	0.130
								Total	0.991	0.979–1.003	0.161
C3M_SF	1.000	0.993–1.006	0.926	0.993		0.947–1.040	0.761	Constant	1.000	0.993–1.008	0.941
								Intermittent	1.001	0.994–1.008	0.847
								Total	1.001	0.003–1.008	0.882
C10C_SF	0.998	1.000–1020	0.718	0.996		0.914–1.084	0.919	Constant	1.005	0.992–1.018	0.497
								Intermittent	1.002	0.989–1.015	0.760
								Total	1.002	0.989–1.016	0.738
ARGS_SF	0.997	0.983–1.006	0.548	1.054		0.977–1.137	0.188	Constant	1.005	0.992–1.018	0.497
								Intermittent	1.006	0.994–1.017	0.347
								Total	1.006	0.993–1.018	0.384

**Table 4 ijms-24-09483-t004:** Biomarkers’ description.

Protein	Biomarker Description	Reflective of	Ref.
Type I collagen	**C1M**: Neo-epitope of MMP-2-, -9-, and -13-mediated degradation of type I collagen	Bone and synovial tissue degradation	[42]
Type II collagen	C2M: Neo-epitope of MMP (multiple)-mediated degradation of type II collagen	Cartilage degradation	[10]
Type III collagen	C3M: Neo-epitope of MMP-9-mediated degradation of type III collagen	Synovial tissue degradation	[43]
Type X collagen	C10C: Cathepsin-K-mediated degradation of type X collagen	Chondrocyte activity	[14]
Aggrecan	ARGS: ADAMTS4/5-mediated degradation of aggrecan	Cartilage degradation	[16]

## Data Availability

The data presented in this study are available upon reasonable request from the corresponding author.

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
