# Peer review of "Relevance of Biomarkers in Serum vs. Synovial Fluid in Patients with Knee Osteoarthritis"

_ijms, 2023, doi:10.3390/ijms24119483_

Round 1

Reviewer 1 Report

The paper is well done and properly developed.Also if the results are not definitive they are extremely useful in order to select the proper essays to better define the OA process.

Reviewer 2 Report

Dear Authors,

I have read with interest Your article entitled "Relevance of biomarkers in serum vs synovial fluid in patients with knee osteoarthritis". In my opinion, the article elaborates on an interesting topic and it is clear and concisely written. Although, I feel some shortcomings in the Introduction section, where between lines 51-80 it seems to me to be only a pile-up of information, fairly known by readers specialized in this field of research. I would recommend reviewing this section by the Authors by shortening these general definitions and completing/focusing mainly on articles that have assessed (even separately) the levels of these biomarkers in the serum and synovial fluid and from this information guide the reader why the present study was relevant to conduct.

Kind regards
